



# Improvements to the use of the Trajectory-Adaptive Multilevel Sampling algorithm for the study of rare events

Pascal Wang[1], Daniele Castellana[2], and Henk A. Dijkstra[2,3]

[1]Univ. Lyon, ENS de Lyon, Univ. Claude Bernard, CNRS, Laboratoire de Physique, Lyon, France
[2]Institute for Marine and Atmospheric research Utrecht, Department of Physics, Utrecht University, Utrecht, The Netherlands.
[3]Centre for Complex Systems Studies, Department of Physics, Utrecht University, Utrecht, The Netherlands.

**Correspondence:** Pascal Wang: pascal.wang@ens-lyon.fr

**Abstract.** The Trajectory-Adaptive Multilevel Sampling (TAMS) is a promising method to determine probabilities of noise induced transition in multi-stable high-dimensional dynamical systems. In this paper, we focus on two improvements of the current algorithm related to (i) the choice of the target set and (ii) the formulation of the score function. In particular, we use confidence ellipsoids determined from linearized dynamics in the choice of the target set. Furthermore, we define a score
function based on empirical transition paths computed at relatively high noise levels. The suggested new TAMS method is applied to two typical problems illustrating the benefits of the modifications.

## 1 Introduction

Systems from various areas of physics exhibit multiple stable states. In such multi-stable systems, transitions between states can occur as a result of small-scale processes, usually referred to as noise-induced transitions (Ashwin et al., 2012). Typical
elements in the Earth's system which show multistability include the Greenland Ice Sheet (Ridley et al., 2010; Robinson et al., 2012), the Amazon Rainforest (Higgins and Scheiter, 2012; Lasslop et al., 2016) and the Atlantic Meridional Overturning Circulation (AMOC). In particular, the latter can undergo transitions to a collapsed state due to fluctuations in the surface freshwater forcing (Castellana et al., 2019).

A central issue in models of these multi-stable systems is the computation of transition probabilities between different states.
If we exclude very special classes of systems, analytical results are generally not available. The Eyring-Kramers formula (Eyring, 1935; Kramers, 1940), which allows the computation of transition rates for reversible processes in the zero noise limit, has been recently generalised to non-gradient systems (Bouchet and Reygner, 2016). However, this method involves the calculation of quasi-potentials, which are generally hard to compute from their variational characterization. From the numerical point of view, the naive method would be following a Monte Carlo approach through performing simulations of large ensembles
of trajectories and calculate transition probabilities by counting the number of trajectories which actually undergo a transition. However, if the occurrence of a transition is a rare event, such computations are not feasible. For instance, to sample an event of probability $p \sim 10^{-8}$, one would need to compute at least $N > N_{min} = 10^8$ trajectories ($N_{min} \sim 1/p$), which is currently impossible to achieve for large-dimensional dynamical systems, where time integrations are expensive.





In order to sample tails of distributions more effectively, various methods have been developed, generally referred to as
rare-event algorithms. One of the promising methods to compute transition probabilities is the Trajectory-Adaptive Multilevel
Sampling (TAMS) method (Lestang et al., 2018). Its underlying idea is to perform a selection/mutation process that discards
trajectories going away from a certain target set and splits/branches from those that get closer to this set. A very similar algo-
rithm (Adaptive Multilevel Splitting or AMS), based on the same approach, has been used in the study of transitions in Jupiter's
turbulent dynamics (Bouchet et al., 2019) and in molecular dynamics to compute the expected dissociation time between a pro-
tein and its ligand (Teo et al., 2016). In these studies, AMS proved to be a powerful tool that reduced computational costs
by several orders of magnitude. Indeed, the required minimum number of computed trajectories scales like $N_{min} \sim 1/\log p$
(Cérou et al., 2016), which is exponentially better than that for the Monte Carlo estimation. The aforementioned selection and
mutation process of discarding and branching trajectories is carried out according to a score function, which allows to rank
trajectories. Rolland and Simonnet (2015) have shown that the choice of the score function plays an important role for the
performance of the algorithm, even for systems with only two degrees of freedom. When using non-optimal score functions,
especially near phase transitions, the variance of the estimated probability can peak and the convergence of the algorithm can
be slow.

The aim of this work is to propose improvements to the use of the TAMS algorithm to be able to compute transitions in
multi-stable systems more efficiently. The first type of improvement is the choice of the target set, which is often determined
from rather arbitrary thresholds. This choice also raises more broadly the question of a precise definition of what we consider a
noise-induced transition between two (stable) states. In the second type of improvement, we propose a more systematic method
of defining a score function, i.e., based on empirical transition paths. The modified TAMS method is first applied to an idealized
gradient system, and then to a system representing a box model of the AMOC (Castellana et al., 2019).

In section 2, we describe the methods developed to improve the TAMS algorithm. In section 3 we show how to incorporate
these techniques into the definition of the score function and present the results for idealised dynamical systems and the AMOC
model. A discussion follows in section 4, assessing the strengths and the limitations of the new TAMS method.

## 2    Methods

### 2.1    Transition probabilities using TAMS

We consider finite-dimensional dynamical systems described by stochastic differential equations (SDEs), of the following
form:

$$\mathrm{d}\mathbf{X}_t = F(\mathbf{X}_t)\,\mathrm{d}t + G\,\mathrm{d}\mathbf{W}_t, \tag{1}$$

where $\mathbf{X}_t \in \mathbb{R}^n$ and $F : \mathbb{R}^n \to \mathbb{R}^n$ is the drift field. The noise term $\mathbf{W}_t \in \mathbb{R}^m$ consists of $m$ independent Wiener processes with
the matrix $G : \mathbb{R}^n \times \mathbb{R}^m$ being the noise matrix.


A prominent example of such a system is a model of a free particle moving in a two-dimensional double-well potential (with

$n = m = 2$). The drift term in the time-evolution equation for the variables $x$ and $y$ is in this case

$$F(x,y) \equiv -\nabla V(x,y) = (x - x^3, -y),  \qquad (2)$$

where $V(x,y)$ represents the potential (Fig. 1(a)). In the deterministic case (i.e. $G = 0$), the stable steady states of the system
are $\mathbf{X}_A = (-1,0)$ and $\mathbf{X}_B = (+1,0)$, while the unstable steady state is $\mathbf{X}_C = (0,0)$. Without fluctuations, a particle starting in
$\mathbf{X}_A$ stays in $\mathbf{X}_A$. However, because of the presence of unresolved processes (such as thermal fluctuations), modelled by the

noise term in the SDE, the particle can move away from $\mathbf{X}_A$ and in some cases make the transition to the state $\mathbf{X}_B$. An example
of such transition is shown in Fig. 1(b).

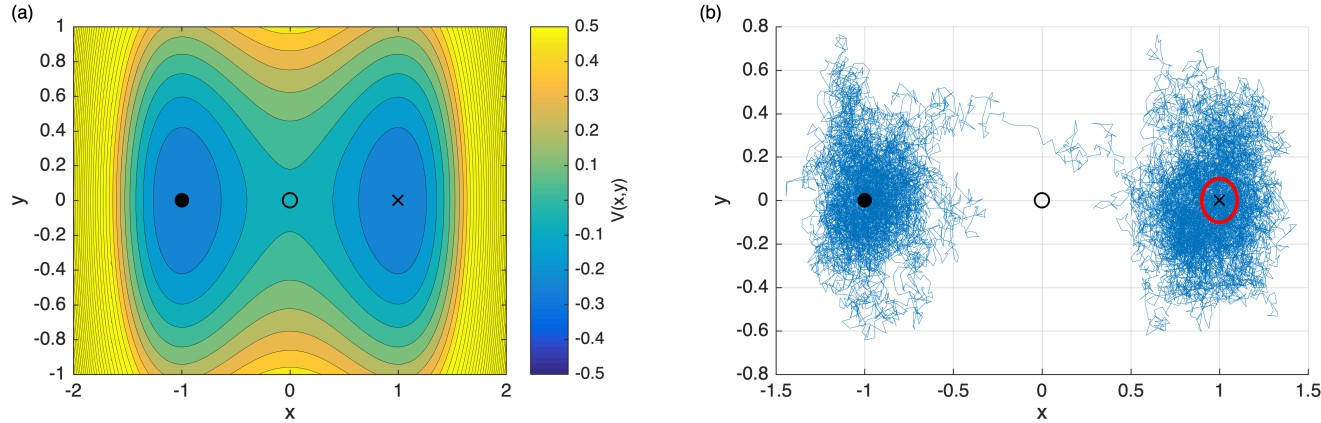

**Figure 1.** (a) Iso-potential contours corresponding to the double-well potential (2). The two stable steady states $\mathbf{X}_A = (-1,0)$ and $\mathbf{X}_B = (+1,0)$ are marked with a filled circle and a cross, respectively. The saddle $\mathbf{X}_C = (0,0)$ is indicated by a circle. (b) Example of a noise-induced transition from $\mathbf{X}_A$ to $\mathbf{X}_B$ for a particle moving in the double-well potential in (a). The noise matrix in the general SDE (1) is chosen as $G = \sigma I$, with $\sigma = 0.32$. The red circle denotes an arbitrarily defined target set $B$.

The transition probability that a trajectory starting in $\mathbf{X}_A$ reaches a neighbourhood $B$ around $\mathbf{X}_B$ before time $T$ is indicated
by $P(\tau_B < T | \mathbf{X}_0 = \mathbf{X}_A)$. Here, $\tau_B$ denotes the stopping time associated with reaching $B$. The TAMS algorithm is based on
a selection and mutation process of discarding and branching trajectories, which are ranked according to a score function

$\phi : \mathbb{R}^n \times \mathbb{R} \to \mathbb{R}$. For a given state $(\mathbf{X}, t) \in \mathbb{R}^n \times \mathbb{R}$, $\phi(\mathbf{X}, t)$ is supposed to measure how likely it is to start in $(\mathbf{X}, t)$ and reach
$B$ before time $T$. As a consequence, if the choice of score function is successful, the probability that a trajectory reaches the
target set $B$ keeps increasing at each iteration of the algorithm, which is why this method is more efficient than brute-force
techniques. A visual representation of the algorithm is given in Fig. 2 and a step-by-step description is provided in Appendix
A (more details can be found in Lestang et al. (2018)).




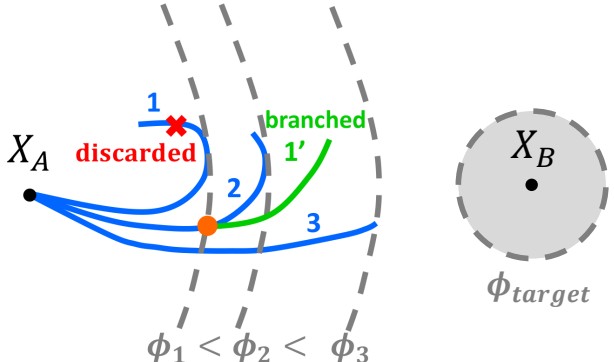

**Figure 2.** Illustration of the TAMS algorithm. First, simulate $N$ trajectories starting in $\mathbf{X}_A$. ($N$=3 blue trajectories in the figure). The trajectories are ranked according to their score $\phi_i$, which is the maximum value of the score function $\phi$ along the trajectory. Then, at each iteration, the trajectory with lowest score (1 in the figure, with score $\phi_1$) is discarded. It is replaced by picking a trajectory uniformly at random from the other ones (2 in the figure), then computing the earliest position at which it reaches a higher score than the discarded trajectory (orange dot) and finally using this position as the branching point for the new trajectory ($1'$). This is repeated until all the trajectories reach $B$ or the number of iterations reaches a predefined limit.

## 2.2 Score functions

Consider a general SDE (1) and two stable states $\mathbf{X}_A$ and $\mathbf{X}_B$ of the corresponding deterministic system. The TAMS algorithm (cf. Appendix A) needs a score function to be defined, which allows to rank trajectories, and select the ones to discard. The optimal score function $\phi_{com}(\mathbf{X}, t)$, i.e. the score function that minimises the variance of the probability estimator, is called the static committor. Its generic expression is given by the following conditional probability (Lestang et al., 2018):

$$\phi_{com}(\mathbf{X}, t) = P(\tau_B < T | \mathbf{X}, t), \tag{3}$$

where $T$ is again the fixed duration of the trajectories and $\tau_B$ is the stopping time associated with reaching the target set $B$. In other words, $\phi_{com}(\mathbf{X}, t)$ is the probability that a trajectory starting in $\mathbf{X}$ at time $t$ reaches $B$ before time $T$. This expression is quite natural because ideally, the score of $(\mathbf{Y}, s)$ should be higher than the score of $(\mathbf{X}, t)$, i.e., $\phi(\mathbf{X}, t) \leq \phi(\mathbf{Y}, s)$, if and only if $P(\tau_B < T | \mathbf{X}, t) \leq P(\tau_B < T | \mathbf{Y}, s)$. This condition is clearly satisfied by $\phi_{com}$ (and in fact any increasing function of $\phi_{com}$). However, the expression given in eq. (3) is generally unusable because it is the very quantity that we want to compute. For instance, $\phi_{com}(\mathbf{X}_A, 0)$ is precisely the transition probability that TAMS estimates. As a conditional probability of the form $P(\mathbf{Y}, s | \mathbf{X}, t)$, the committor $\phi_{com}(\mathbf{X}, t)$ satisfies the backward Fokker Planck equation (Lestang et al., 2018):

$$\frac{\partial \phi}{\partial t} + F \cdot \nabla \phi + \frac{1}{2} \sum_{i,j} G_{ij} \frac{\partial^2 \phi}{\partial X_i \partial X_j} = 0, \quad \text{with boundary conditions} \quad \begin{cases} \forall \mathbf{X} \in \partial B, \forall t \in [0, T], \phi(\mathbf{X}, t) = 1, \\ \forall \mathbf{X} \in \mathbb{R}^n \setminus B, \phi(\mathbf{X}, T) = 0. \end{cases} \tag{4}$$





However, solving the backward Fokker Plank equation in systems with many degrees of freedom is computationally infeasible.

Moreover, even if the committor is available on a grid used for the discretization of (4), using interpolation to evaluate it during a TAMS loop can also have a prohibitive computational cost.

Bouchet et al. (2019) made use of a score function based on the distances of the state $\mathbf{X}$ from the starting state $\mathbf{X}_A$ and the destination equilibrium $\mathbf{X}_B$, respectively: it is defined as

$$\phi_{dist}(\mathbf{X}) \equiv \begin{cases} d_A/2d_B & \text{if } d_A < d_B \\ 1 - d_B/2d_A & \text{otherwise} \end{cases} \quad \text{with } d_A \equiv \|\mathbf{X} - \mathbf{X}_A\|, d_B \equiv \|\mathbf{X} - \mathbf{X}_B\| \tag{5}$$

Alternatively, a Gaussian-shaped score function $\phi_{gauss}$ was proposed in Baars (2019). It is defined as:

$$\phi_{gauss}(\mathbf{X}) \equiv \eta - \eta e^{-\beta\|\mathbf{X}-\mathbf{X}_A\|^2/\|\mathbf{X}_C-\mathbf{X}_A\|^2} + (1-\eta)e^{-\beta\|\mathbf{X}-\mathbf{X}_B\|^2/\|\mathbf{X}_C-\mathbf{X}_B\|^2} \tag{6}$$

with $\eta \equiv \|\mathbf{X}_C - \mathbf{X}_A\|/\|\mathbf{X}_A - \mathbf{X}_B\|$. Here $\beta \in \mathbb{R}$ is a parameter controlling the decay and $\mathbf{X}_C$ is the saddle state of the system. Due to the general expressions of $\phi_{gauss}$ and $\phi_{dist}$, these score functions can be used in systems of any dimension.

### 2.3 Definition of the target set

Once the score function has been chosen, a threshold needs to be defined for the TAMS algorithm to converge, so that the occurrence of a transition can be detected. In other words, we do not expect each trajectory that undergoes a transition to reach exactly the destination equilibrium $\mathbf{X}_B$, but rather a neighbourhood of it ($B$). The target set $B$ can then be defined according to a level set $\phi^{target}$ of the score function $\phi$:

$$B = \{\mathbf{X} \in \mathbb{R}^n \mid \phi(\mathbf{X}) > \phi^{target}\} \tag{7}$$

However, different score functions $\phi$ and different level sets $\phi^{target}$ correspond to different target sets $B$, which can differ in volume and in shape. Moreover, often the level set $\phi^{target}$ is defined somewhat arbitrarily. For example, by using $\phi_{gauss}$ with target score $\phi^{target} = 0.85, 0.9$ or $0.95$, we found that the average of the transition probability estimator for a two-dimensional double-well potential system (2) can vary up to 30%. This may not be a concern if one only cares about the order of magnitude of the transition probability but it will be problematic if quantitative comparisons are needed. Moreover, a poor choice of target

set can lead to inaccuracies when a trajectory has a score greater than $\phi^{target}$ without actually making the transition in all the degrees of freedom. Thus, there is a need for defining a canonical choice of a target set $B$, which needs to fundamentally address what is considered to be a noise-induced transition.

For this purpose, we use the concept of confidence ellipsoid. This is an ellipsoidal neighbourhood around a stable equilibrium state, inside which a trajectory subject to the locally linearized dynamics stays, within a certain confidence level (Cowan,

1998). Consider a general SDE system given by (1). Because the drift vanishes at an equilibrium state $F(\mathbf{X}_B) = 0$, its first order approximation around the equilibrium state $\mathbf{X}_B$ via Taylor expansion is:

$$F(\mathbf{X}) = A(\mathbf{X}_B)(\mathbf{X} - \mathbf{X}_B) + \mathcal{O}(\|\mathbf{X} - \mathbf{X}_B\|^2), \tag{8}$$





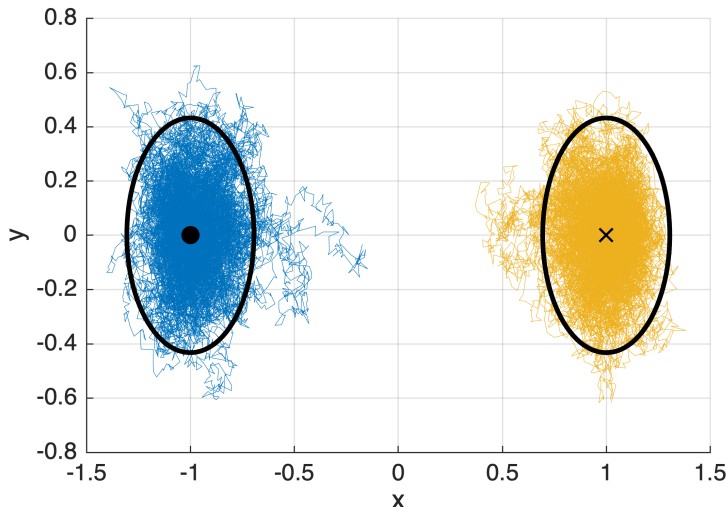

**Figure 3.** Confidence ellipsoids for the two-dimensional double-well potential system (2), with confidence level $1 - \alpha = 0.95$ and noise matrix $G = 0.25I$, where $I$ is the identity matrix in $\mathbb{R}^2$. Two trajectories of duration $T = 200$ are shown: one (blue) is initialised at the initial state and one (orange) at the target state. 93% of the points composing a trajectory are inside their corresponding ellipsoid. This is lower than the presribed confidence level because away from the equilibrium, the first order dynamics from which the confidence ellipsoids are derived does not hold.

where $A(\mathbf{X}_B) = \nabla F(\mathbf{X}_B) \in \mathbb{R}^{n \times n}$ is the Jacobian matrix of $F$ at $\mathbf{X}_B$. Using a translation $\widetilde{\mathbf{X}} = \mathbf{X} - \mathbf{X}_B$, the first order approximation of the SDE is then:

$$\mathrm{d}\widetilde{\mathbf{X}}_t = A(\mathbf{X}_B)\widetilde{\mathbf{X}}_t \mathrm{d}t + G\,\mathrm{d}\mathbf{W}_t. \tag{9}$$

Because the drift term has been linearized, this is the equation for an $n$-dimensional Ornstein-Uhlenbeck process. The stationary probability density function (PDF) $f$ of the approximating process is Gaussian (Cowan, 1998) and given by

$$f(\mathbf{X}) = \frac{1}{(2\pi)^{\frac{n}{2}} |C_B|^{\frac{1}{2}}} e^{-\frac{1}{2} \|\mathbf{X} - \mathbf{X}_B\|^2_{C_B^{-1}}}, \tag{10}$$

where $C_B \in \mathbb{R}^{n \times n}$ is the covariance matrix of the system calculated in $\mathbf{X}_B$ and $\|.\|_{C_B^{-1}}$ the norm induced by its inverse

$C_B^{-1}$, defined by $\|\mathbf{X}\|^2_{C_B^{-1}} \equiv \mathbf{X}^\top C_B^{-1} \mathbf{X}$. The covariance matrix $C_B$ can be thought heuristically as the matrix containing the correlations $\mathbb{E}(x_i x_j)$ (with $\mathbf{X} = (x_1, \ldots, x_n)$), which generalises the notion of variance in n-dimensions. $C_B$ is obtained by solving the Lyapunov equation (see Kuehn, 2012, for the full derivation):

$$A(\mathbf{X}_B)C_B + C_B A(\mathbf{X}_B)^\top + GG^\top = 0 \tag{11}$$

We then define the confidence ellipsoid, which has $C_B^{-1}$ as shape matrix, as follows

$\mathcal{E} = \{\mathbf{X} \in \mathbb{R}^n \mid \|\mathbf{X} - \mathbf{X}_B\|^2_{C_B^{-1}} \equiv (\mathbf{X} - \mathbf{X}_B)^\top C_B^{-1}(\mathbf{X} - \mathbf{X}_B) < Q_\alpha\}, \tag{12}$





where $Q_\alpha$ is the quantile of confidence level $1 - \alpha$ of the $n$-dimensional $\chi^2$ distribution (Cowan, 1998); usually $1 - \alpha = 0.95$. The directions of symmetry of the ellipsoids are given by the eigenvectors of the covariance matrix $C_B$ and the radii are given by the corresponding eigenvalues and the confidence level $1 - \alpha$. Intuitively, the greater the eigenvalue, the more a trajectory fluctuates in the given direction.

The $(1 - \alpha)$-covariance ellipsoid represents the $n$-dimensional volume where a trajectory is confined with confidence level $1 - \alpha$, provided its dynamics is well approximated by the first order expansion at the equilibrium point. An illustration of the 0.95-confidence ellipsoid for the double-well potential is shown in Fig. 3. The confidence ellipsoid $\mathcal{E}$ constitutes a way to meaningfully define the target set $B$ with minimal arbitrary parameters. As shown in Fig. 3, when initializing a trajectory at $\mathbf{X}_A$ or $\mathbf{X}_B$ in the two-dimensional double-well system, it stays inside the correspondent ellipsoid with a certain confidence

$1 - \alpha = 0.95$. In the next section, we show how to incorporate this choice of target set in the score function $\phi$.

### 2.4 Estimating the typical transition path using histograms

The second line of improvement of the score function concerns the estimation of typical transition paths of the dynamical system. In the zero noise limit, the Freidlin-Wentzell theory of large deviations predicts that transition paths cluster around the most probable transition path, called the instanton (Freidlin and Wentzell, 1984). On the other hand, in the finite noise regime,

transition paths may deviate from the instanton. Moreover, instantons may be computationally inaccessible for systems with many degrees of freedom. Therefore, it can prove more relevant to estimate empirically the typical transition path that the system follows at a given finite noise level, which is the approach we follow here.

The idea is to first accumulate transition paths at a noise level where transitions are frequent enough (typically $p > 10^{-3}$) so that any sampling method (direct Monte Carlo or TAMS with naive score functions) can be used. Then, we compute

the spatial histogram of the transition paths over a discretized phase space using $n-$dimensional boxes. This provides the spatial distribution of the transition paths, which is concentrated around a typical transition path, reminiscent of an instanton phenomenology, which was also observed in more complex systems (Bouchet et al., 2019). From the spatial histogram, we extract a typical transition path. The main steps of the path-finding algorithm are listed below:

(i) the trajectory of the typical transition path starts in the box of the histogram containing the initial state $\mathbf{X}_A$;

(ii) the next box in this trajectory corresponds to the neighbour which has the highest nonzero histogram value but which has not already been visited by the typical transition path;

(iii) the algorithm stops if it reaches the box containing the target state $\mathbf{X}_B$.

In addition, the full typical path estimation algorithm uses a self-correcting method to avoid dead ends when there are no valid neighbours to be the next point in the trajectory. The spirit of the path-finding algorithm is similar to the depth-first search

algorithm (Cormen et al., 2009). We found that, as long as the histogram is not fragmented, i.e., there is a sufficient number of accumulated trajectories or large enough histogram bins, the algorithm converges. Possible artefacts created by this estimation include spiralling near the initial equilibrium (because of the concentric shell structure of the local probability density function



(10) and zigzagging at the histogram box size. They can be both addressed by a clean-up algorithm: starting from the first box, at each box $\mathbf{X}_j$, if the trajectory goes back to one of its neighbours at a later time, with $\mathbf{X}_{j+k}$ being the latest neighbour visit,

we erase the points $\mathbf{X}_{j+1}, \ldots, \mathbf{X}_{j+k}$ from the trajectory.

## 3 Results

In this section, we apply both modifications to TAMS (ellipsoids in the score function and typical path estimation) to different problems.

### 3.1 Incorporating ellipsoids in the score function

First, we apply the modified TAMS method to the two-dimensional double-well system defined by (1). Here we see, which will be more general, that the level sets of the score function do not have the shape of an ellipsoid. Hence, there is no level $\phi^{target}$ such that the 0.95-confidence ellipsoid $\mathcal{E}$ coincides with $B = \{\mathbf{X} \in \mathbb{R}^n \mid \phi(\mathbf{X}) > \phi^{target}\}$. Here, we propose a general method to modify any target score function so that we are able to choose the target set to be exactly the confidence ellipsoid of $\mathbf{X}_B$. Let

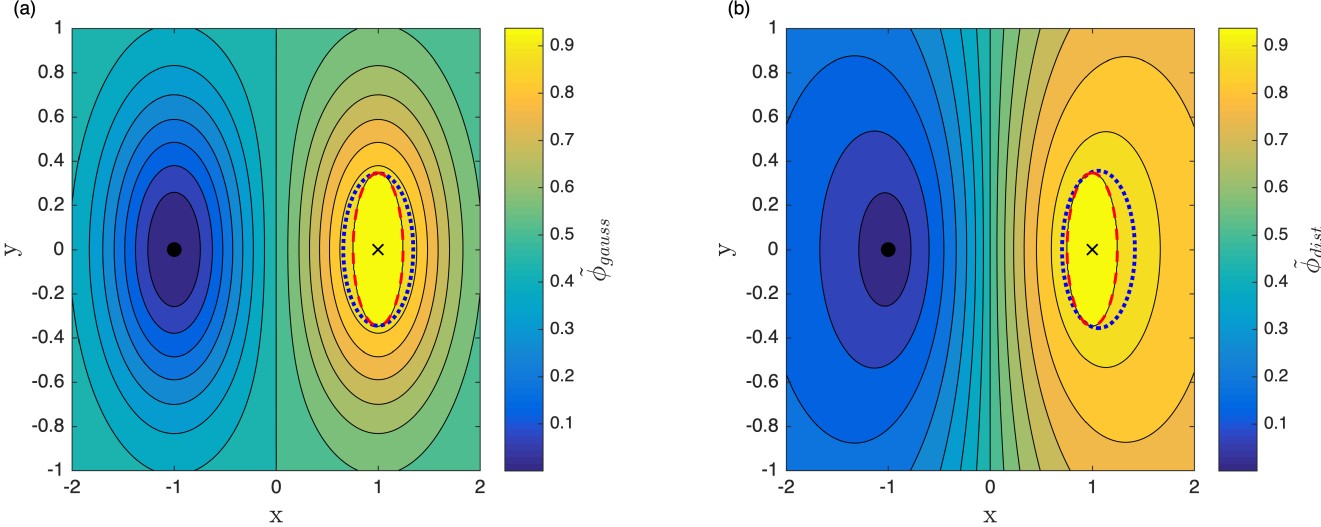

**Figure 4.** (a) Contour levels of the modified score function $\widetilde{\phi}_{gauss}$, for the two-dimensional double-well system represented in (1), according to the procedure described in eq. 14. The level set $\phi^{target}$ (dotted blue line) of the former score function $\phi_{gauss}$ is tangent to the ellipsoid. The level set $\widetilde{\phi}^{target}$ (dotted red) of the modified score function $\widetilde{\phi}_{gauss}$ coincides with the ellipsoid. (b) Same plot, for the score function $\widetilde{\phi}_{dist}$.

$\mathcal{E}$ be the 0.95-confidence ellipsoid around the equilibrium state $\mathbf{X}_B$ and $\phi$ (e.g. $\phi_{gauss}$) the score function to be modified. We

first compute the level $\widetilde{\phi}^{target}$, defined as the minimum of the score function $\phi$ on the confidence ellipsoid $\mathcal{E}$:

$$\widetilde{\phi}^{target} \equiv \min_{\mathbf{X} \in \mathcal{E}} \phi(\mathbf{X}) \tag{13}$$




such that the set $\{\mathbf{X} \in \mathbb{R}^n \mid \phi(\mathbf{X}) > \widetilde{\phi}^{target}\}$ contains the ellipsoid $\mathcal{E}$ and is tangent to $\mathcal{E}$. This can be done numerically by generating a mesh of points around $\mathbf{X}_B$, then selecting the points inside $\mathcal{E}$ by comparing their norm $\|\mathbf{X} - \mathbf{X}_B\|^2_{C_B^{-1}}$ with the quantile $Q_\alpha$ and finally computing the minimum $\widetilde{\phi}^{target}$ of $\phi$ on these points. Then, define the modified score function $\widetilde{\phi}$ in the following way:

$$
\widetilde{\phi}(\mathbf{X}) \equiv \begin{cases} 1 & \text{if } \mathbf{X} \in \mathcal{E} \text{ i.e. } \|\mathbf{X} - \mathbf{X}_B\|^2_{C_B^{-1}} < Q_\alpha \\ \widetilde{\phi}^{target} & \text{if } \phi(\mathbf{X}) > \widetilde{\phi}^{target} \text{ and } \mathbf{X} \notin \mathcal{E} \\ \phi(\mathbf{X}) & \text{otherwise} \end{cases} \tag{14}
$$

The target set $B = \{\mathbf{X} \in \mathbb{R}^n \mid \widetilde{\phi}(\mathbf{X}) > \widetilde{\phi}^{target}\}$ for the modified score function $\widetilde{\phi}$ turns out to coincide with $\mathcal{E}$. We apply this procedure on both the score functions $\phi_{gauss}$ and $\phi_{dist}$ and the results for the improved score functions $\widetilde{\phi}_{dist}$ and $\widetilde{\phi}_{gauss}$ are shown in Fig. 4.

## 3.2 Designing a score function based on a typical transition path

In order to show how to design a score function based on a typical transition path, we consider a two-dimensional system slightly less trivial than the double-well system, i.e. a two-dimensional system with the following potential:

$$
V(x,y) = \underbrace{0.1x^2 + 0.05y^2}_{\text{global confinement}} + \underbrace{30e^{-(x/2)^2}(1 + \tanh(15 - y))}_{\text{potential barrier at } y<15,\, x=0} +
$$

$$
\underbrace{-10e^{((x+6)/2)^2 - (y/2)^2}}_{\text{left potential well}} \underbrace{-10e^{-((x-6)/2)^2 - (y/2)^2}}_{\text{right potential well}}, \tag{15}
$$

depicted in Fig. 5(a). It consists in two energy minima at $\mathbf{X}_A \approx (-5.77, 0)$ and $\mathbf{X}_B \approx (5.77, 0)$ and a potential barrier spanning $y < 15$ and at $x = 0$. The dynamics is then given by the SDE (1), with drift $F(\mathbf{X}_t) = -\nabla V(\mathbf{X}_t)$.

The dynamics of the system is quite interesting, as it exhibits two distinct regimes for transition paths, depending on the noise level $\sigma$ (assuming $G = \sigma I$). At high noise ($\sigma^2 \gg \Delta V$, where the potential barrier height is $\Delta V = V(0,0) - V_A$), trajectories are likely to cross the potential barrier. At low noise ($\sigma^2 \ll \Delta V$), trajectories are not likely to cross the barrier and the trajectories which undergo the transition instead go through the upper channel at $y \sim 15$. Typical examples of such trajectories are shown in Fig. 5(b). Rolland and Simonnet (2015) investigated the convergence properties of AMS using a triple-well potential system, which also exhibits two regimes of preferred transition paths depending on the noise. They found a strong dependency of the statistics of the rare-event algorithm (e.g., the number of iterations) and the duration of reactive trajectories on the choice of score function. We expect the same behaviour when applying TAMS to the system with potential (15) and that this system reveals differences in performance between various score functions.

Fig. 6(a) shows an histogram computed with 300 transition paths for the system with the potential given by eq. (15), with noise level set at $\sigma = 3$. On top of it, a typical transition path was estimated using the algorithm sketched in section 2.4. As


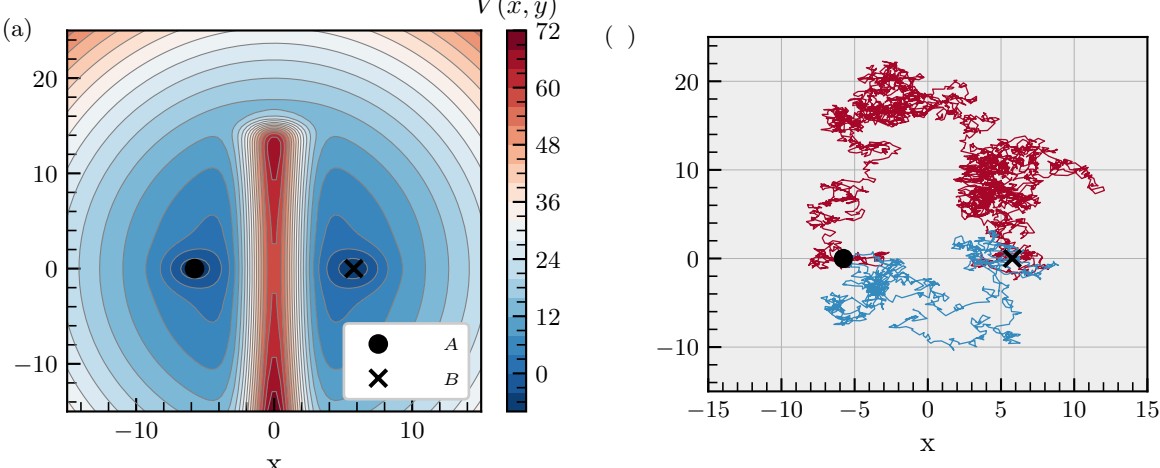

**Figure 5.** (a) Potential landscape of the two-dimensional gradient system defined by eq. (15). Two energy minima are located at $\mathbf{X}_A \approx (-5.77, 0)$ and $\mathbf{X}_B \approx (5.77, 0)$. They are separated by a potential barrier spanning $y < 15$ and at $x = 0$. (b) Typical transition paths from $\mathbf{X}_A$ to $\mathbf{X}_B$ for two noise levels $\sigma$. At high noise, trajectories can cross the potential barrier ($\sigma = 10$, blue). At low noise, trajectories go through the upper channel ($\sigma = 3$, red).

already mentioned, some artefacts created by the estimation (such as spiraling or zigzagging) can be corrected using a clean-up algorithm. The result is shown in Fig. 6(b): the empirical estimation of the typical path resembles the instanton around which the transition paths are clustered at lower noise (Fig. 6(b)). The instanton was calculated by implementing the geometric
minimum action method (Heymann and Vanden-Eijnden, 2008). Note that if, instead, we accumulated trajectories at high noise $\sigma > 10$, we would obtain trajectories going from $\mathbf{X}_A$ to $\mathbf{X}_B$ in a straight line, which is the typical path at high noise similar to Rolland and Simonnet (2015). This typical transition path is then substantially different from the instanton, which goes through the upper channel. Thus, our method can be advantageous in multistable systems where the typical transition path depends on the noise level. We can start at high noise and reapply the empirical estimation of the typical path each time the noise level is
decreased.

Given a typical transition path $\mathcal{C}$, we present the design of a score function $\phi_\mathcal{C}$ which encourages trajectories to follow the transition path $\mathcal{C}$ such that it gives a reasonable approximation of the static committor. Let us consider a trajectory $\mathcal{C}(s)$ in $\mathbb{R}^n$ parametrised by arclength $s \in [0, 1]$. Then we define the score function $\phi_\mathcal{C}$, called the path-based score function, such that it grows from 0 to 1 along the trajectory from $\mathbf{X}_A$ to $\mathbf{X}_B$ and decays exponentially along the direction transverse to the trajectory:

$\phi_\mathcal{C} : \mathbb{R}^n \to [0, 1]$

$$\mathbf{X} \mapsto s(\mathbf{X}, \mathcal{C}) \exp\left(-\frac{d(\mathbf{X}, \mathcal{C})^2}{d_0^2}\right) \tag{16}$$

where $d(\mathbf{X}, \mathcal{C}) = \inf_{s \in [0,1]} \|\mathbf{X} - \mathcal{C}(s)\|_d$ is the distance between $\mathbf{X}$ and the trajectory $\mathcal{C}(s)$, $s(\mathbf{X}, \mathcal{C})$ is the curvilinear coordinate of the position on the trajectory that satisfies the infimum in the definition of $d(\mathbf{X}, \mathcal{C})$ and $d_0$ is the characteristic decay length


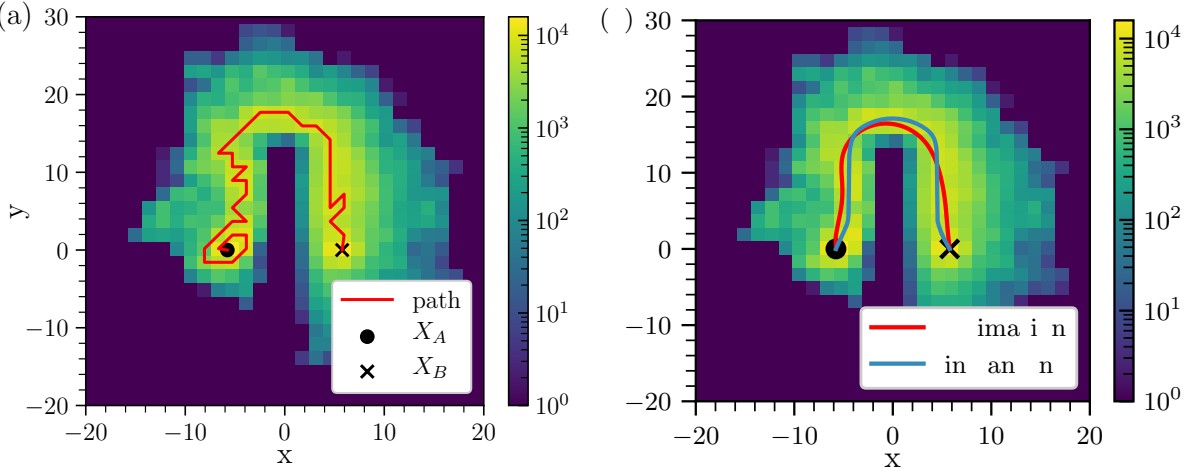

**Figure 6.** (a) Histogram of $N = 300$ transition paths at noise level $\sigma = 3$, using the modified score function $\widetilde{\phi}_{gauss}$ with $\beta = 1.5$, defined by eq. (6) and implemented with (14). The corresponding transition probability ($p = 2 \times 10^{-3}$) is high enough so that Direct Monte Carlo sampling could have been used to produce a similar histogram. The bin resolution ($\Delta x = 1.4$, $\Delta y = 1.75$) is coarse for illustration purposes. The histogram is used as input in the path-finding algorithm which produces the transition path in red. Grid-scale spiralling occurs near the initial state $\mathbf{X}_A$ because of the concentric shell structure of the local probability density function given by eq. (10). (b) Same histogram as the left panel. The estimated typical transition path (red) has been cleaned up from its grid-scale spiraling and zigzags with the clean-up algorithm and has then been smoothed. It strongly resembles the instanton (blue) which was computed by implementing the geometric action minimum method (Heymann and Vanden-Eijnden, 2008).

(free parameter). The score function $\phi_{\mathcal{C}}$ is shown in Fig. 7 for the estimated transition path $\mathcal{C}$ shown in Fig. 6 and two values

of decay length $d_0 = 20$ and $d_0 = 200$. The score function increases from 0 to 1 along the trajectory. Thus, it encodes the preferred direction that the system has to follow. This contrasts with the generic score functions $\phi_{gauss}$ and $\phi_{dist}$ which are symmetrical in $y$: they do not contain the information that the system has to increase in $y$ in order to make the transition to $\mathbf{X}_B$. Note that the method we developed here can be applied, in principle, to systems of any dimension. As shown in Fig. 7(b) the score function $\phi_{\mathcal{C}}$ is discontinuous because the trajectory has positive curvature. The discontinuity is located near the axis

$x = 0$. Indeed, when crossing the axis $x = 0$, the closest point on $\mathcal{C}$ changes and $s(\mathbf{X}, \mathcal{C})$ is discontinuous.

Next, we applied TAMS with the path-based score function $\phi_{\mathcal{C}}$ to the two-dimensional system (15). We compare its performance with the previously defined score functions $\phi_{dist}$ and $\phi_{gauss}$. In fact, we use the associated modified score functions, such that the target set $B$ matches the 0.95-confidence ellipsoid (we drop the tildes for readability). We use the following parameters: duration of a trajectory $T = 20$, time step $\mathrm{d}t = 0.01$. We show in Fig. 8(a) the transition probability estimates using

the score functions $\phi_{dist}$, $\phi_{gauss}$ (with $\beta = 1.5$) and $\phi_{\mathcal{C}}$ (with decay parameters $d_0 = 2, 20, 200$) averaged over 10 instances of the algorithm. The probability estimates are in good agreement between each other and with a Monte Carlo estimation for $\sigma > 2.5$. The score function $\phi_{\mathcal{C}}$ is robust with respect to the choice of the decay length $d_0$.

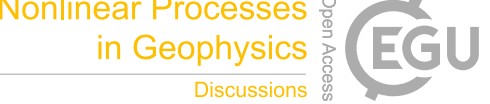

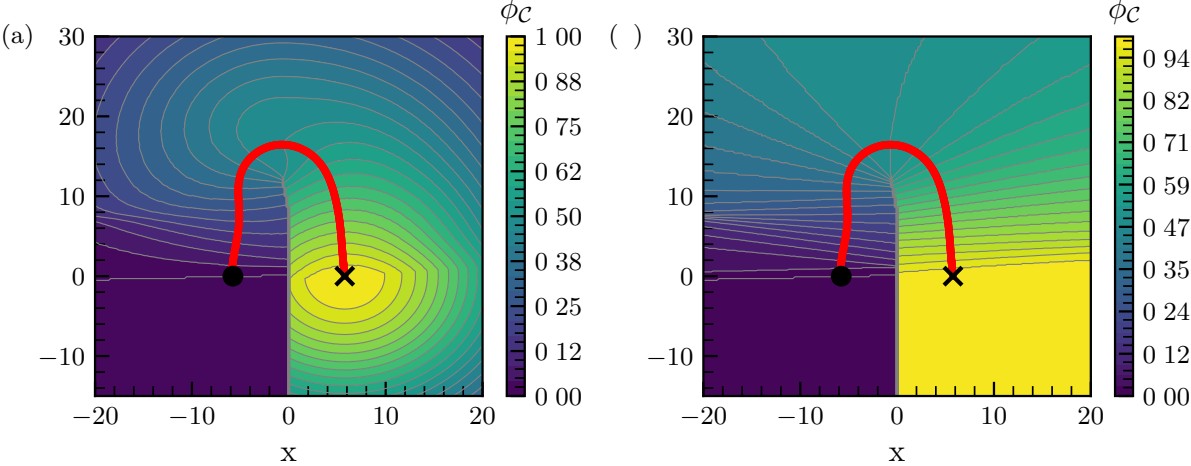

**Figure 7.** (a) Contour levels of the score function $\phi_{\mathcal{C}}$ defined in eq. (16), associated with the estimated transition path $\mathcal{C}$ shown in Fig. 6 and with decay length $d_0 = 20$. (b) Same figure for $d_0 = 200$.

The performances of the numerical methods are next measured using the work-normalised relative error $\epsilon$, which combines the variance of the algorithm and its computational cost (Glynn et al., 2009):

$$\epsilon \equiv \frac{\sigma_{\hat{p}}}{\mu_{\hat{p}}} \times \omega, \tag{17}$$

where $\mu_{\hat{p}}$ and $\sigma_{\hat{p}}$ are the mean and standard deviation of the probability estimate over the different instances, $\omega$ is the average number of time steps calculated in one realisation. In short, $\epsilon$ measures how precise the numerical method is at equal computational cost. The smaller $\epsilon$, the better the algorithm performs.

The results are shown in Fig. 8(b). For the system (15), the score functions $\phi_{dist}$, $\phi_{gauss}$ and $\phi_{\mathcal{C}}$ have little difference in performance. For the lowest noise values, the path-based score function has at most a 30% smaller error $\epsilon$ than the score function $\phi_{gauss}$. Changing the decay length $d_0$ hardly changes the error $\epsilon$. When adjusting the parameters of the potential $V$ or applying the same method to the triple well system used in Rolland and Simonnet (2015), the performance gain, while being often present, never systematically exceeded 30%. All in all, in this category of two-dimensional systems, using the path-based score function approach yields little improvement.

## 3.3 Transition probabilities in a box model of the AMOC

Finally, as a main application of one of the techniques shown in this paper, we consider the dynamical system in Castellana et al. (2019), which represents a box model of the Atlantic Meridional Ocean Circulation (AMOC). The system consists of a set of stochastic differential equations, plus one algebraic constraint, the latter representing the salt conservation in the model

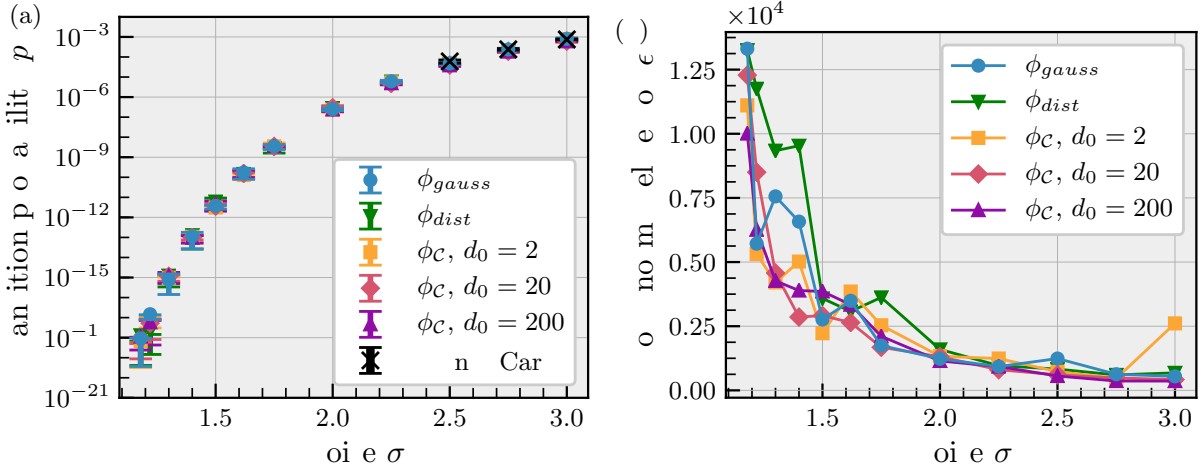

**Figure 8.** (a) Transition probability $p$ as a function of noise parameter $\sigma$. Mean estimates and interquartile range (error bars) over $N_{samples} = 10$ independent realisations of the TAMS algorithm using the score function $\phi_{dist}$ (green down triangle) $\phi_{gauss}$ (with $\beta = 1.5$, blue circle) and $\phi_{\mathcal{C}}$ with decay parameters $d_0 = 2, 20, 200$ (yellow square, pink diamond, purple up triangle) are shown. The Monte Carlo estimate (black cross) has been run with $N = 5 \times 10^5$ trajectories with target set the confidence ellipsoids around $\mathbf{X}_B$, with associated standard deviation $\sigma_p = \sqrt{p(1-p)/N}$ (error bars). There is an overall good agreement between the numerical estimations. The path-based score function is robust to the choice of decay parameter. (b) Performance of the score functions measured by the work-normalised error $\epsilon$ as a function of noise $\sigma$ (same markers as left panel). At most, there is a 30% decrease in error when using the path-based score function $\phi_{\mathcal{C}}$ at low noise. Note that at high noise $\sigma = 3$, using a short decay length $d_0 = 2$ with $\phi_{\mathcal{C}}$ leads to poor performance because the greater values of $\phi_{\mathcal{C}}$ are tightly concentrated around the estimated instanton whereas typical transition paths are not necessarily clustered around it. Otherwise, the performance of the score functions are roughly similar.

(as shown in Appendix B). This model can be formulated as

$$\mathrm{d}\mathbf{Y}_t = F_1(\mathbf{Y}_t, Z_t)\,\mathrm{d}t + G\,\mathrm{d}\mathbf{W}_t$$
$$0 = F_2(\mathbf{Y}_t, Z_t) \tag{18}$$


In the equations above, we split the state of the system $\mathbf{X}_t$ into two parts: $\mathbf{Y}_t$, which includes salinities of four of the boxes plus the depth of the pycnocline $D$, and $Z_t$, which represents the salinity of the deep box ($S_d$). As the noise is applied only on the asymmetric component of the atmospheric freshwater flux ($E_a$), it directly affects only two of the variables $S_n$ and $S_s$). Moreover, the stochastic increments associated with the two variables are identical, to make sure that each decrease of

freshwater forcing in the southern box results in the same increase of it in the northern box in the model. Therefore, the noise is not spatially independent and the noise matrix $G$ is no longer diagonal: it consists of a $(5 \times 1)$ row vector, with only two elements different from zero. For a reasonable choice of the parameters, the deterministic system is in a bistable regime (Castellana et al., 2019), which means that there are two possible equilibrium states under the same forcing conditions. In general, we are interested in studying transitions between the present-day AMOC ($\mathbf{X}_A$) and the collapsed state ($\mathbf{X}_B$).





For a differential-algebraic system of equations (DAEs), such as the system in eq. (18), we need to be particularly careful while computing the covariance ellipsoids. First of all, we make use of the Schur complement of the Jacobian of the system, which allows to calculate the covariance matrix when an algebraic constraint is present (Baars et al., 2017). Nevertheless, the resulting matrix $C_B$ is singular, with two eigenvalues being equal to zero. One of the corresponding eigenvectors is a vector pointing in the direction of the variable $D$ (depth of the pycnocline). The reason behind it is that the differential equation

governing the evolution of $D$ does not contain any of the other variables (see Castellana et al., 2019), when the system is in the collapsed state ($\mathbf{X}_B$). This results in $D$ not being affected by the noise, as this is imposed only on two of the salinities. Hence, we compute the covariance matrix $C_B^S$ relative to the salinities, removing one degree of freedom from the original matrix. Unfortunately, such matrix still has one zero eigenvalue, which is due to the salinity conservation (the algebraic equation in the system (18)). To overcome this problem, we compute the Moore-Penrose inverse (or pseudoinverse) of the covariance matrix,

$C_B^{S+}$, by performing a singular value decomposition of $C_B^S$ and removing the zero eigenvalue, together with the corresponding eigenvector (Ben-Israel and Greville, 2003). A two-dimensional projection of the ellipsoid constructed for the box model is shown in Fig. 9.

       For the system (18), the modified score function is more complicated, as the covariance matrix used to construct the ellipsoid contains only the degrees of freedom related to the salinities of the model, leaving the variable $D$ (depth of the pycnocline)

out. From a geometric point of view, that means that the covariance ellipsoid around $\mathbf{X}_B$ is degenerated along the $D$-direction. Fig. 9 shows a projection of the ellipsoid - once the noise amplitude is fixed - on the plane identified by the variables $S_n$ and $S_s$ (respectively, the salinity of the northern box and the one of the southern box in the model). The projection was obtained calculating the conditional covariance matrix of the two variables into consideration, given that the other variables are set on their mean value (Wasserman, 2013). The two-dimensional ellipse contains the large majority of the projected time points (on

the same plane) of a trajectory that wanders around the equilibrium. By construction, the confidence level of the confinement is higher than the one prescribed for the full dimensional ellipsoid (in this case 0.95). Clearly the level sets of the score function $\phi_{gauss}$ do not coincide with the one of the ellipsoid. Hence, the importance of modifying the score function appears evident. When constructing an improved score function, in order to evaluate if a state belongs to the neighbourhood of $\mathbf{X}_B$, we need to check two conditions: (i) whether the state of the system is inside the salinity covariance ellipsoid drawn around the destination

equilibrium, and (ii) we need to verify that the variable $D$ of the state is the same as the one of $\mathbf{X}_B$. Hence, the improved score function for the box model reads

$$\widetilde{\phi}_{box}(\mathbf{X}) \equiv \begin{cases} 1 & \text{if } \mathbf{X}^S \in \mathcal{E}^S \text{ and } X^D = X_B^D \\ \widetilde{\phi}^{target} & \text{if } \phi_{gauss}(\mathbf{X}) > \widetilde{\phi}^{target} \text{ and } (\mathbf{X}^S \notin \mathcal{E}^S \text{or } X^D \neq X_B^D) \\ \phi_{gauss}(\mathbf{X}) & \text{otherwise} \end{cases} \quad (19)$$

where $\mathbf{X}^S$ indicates the part of the state vector representing the set of the salinities, whereas $X^D$ represents the variable $D$. As already mentioned, to check whether a certain state belongs to the salinity ellipsoid, we made use of the pseudo-inverse of $C_B^S$

in the definition (12).

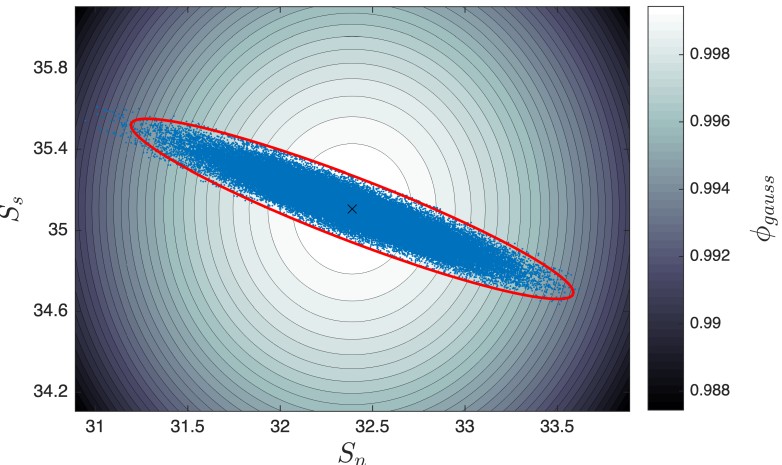

**Figure 9.** Level sets of the score function $\phi_{gauss}$, in proximity of the collapsed equilibrium state $\mathbf{X}_B$ of the system in Castellana et al. (2019), projected on the plane identified by the variables $S_n$ and $S_s$ (respectively, the salinity of the northern box and the one of the southern box in the model). A trajectory of the system, initiated in the collapsed state, has been projected on the same plane (blue circles). The red ellipse is the two-dimensional projection of the covariance ellipsoid constructed by using the matrix $C_B^S$ and a confidence level of 0.95; 98% of the points composing the trajectory are inside the ellipsoid.

To be able to assess the relevance of a proper definition of the target set in the TAMS algorithm, and hence the importance of using the improved version of the score function, we computed transition probabilities of the AMOC from the present-climate state to the collapsed state for reasonable values of the atmospheric forcing and noise. In particular, we chose $\bar{E}_a = 0.20\,Sv$ and $f_\sigma = 0.16$. This last value represents the ratio between the standard deviation of the noise in the atmospheric forcing and its mean value (Castellana et al., 2019). The number of trajectories used in the algorithm was set to 100, and the time scale at which the probabilities were evaluated was chosen to be $1,000$ and $10,000$ years, respectively. For each transition probability, we used three different versions of the algorithm, based on three different settings for the score function. The first two versions were implemented using $\phi_{gauss}$ in (6), with two different choices for the threshold $\phi^{target}$. In the third version, we used $\widetilde{\phi}_{\text{box}}$ as defined in (19), where the starting score function was $\phi_{gauss}$. Each probability was calculated running 15 instances of the algorithm, and then computing the median and the interquartile range (IQR). The results are shown in the Table 1 below.

When running TAMS to compute transition probabilities between two states of the Atlantic Circulation in this model, with different versions of the score function, we found a considerable discrepancy between the obtained values. In particular, it appears that setting a very high threshold in the Gaussian score function makes the algorithm detect no transitions (we set up the algorithm so that it stops when the probabilities involved are smaller than $10^{-9}$): the reason behind this is that, because of the presence of the noise, we don't expect the state of the system to stay indefinitely close to the destination equilibrium, but rather to wander around it. Therefore, the score function, which assigns the maximum score only to a very small neighbourhood of the equilibrium, is not able to properly recognise transitions. Moreover, it is not surprising that, when using $\phi_{gauss}$ with a





| Score function | $p$ at 1000 years [IQR] | $p$ at 10000 years [IQR] |
|---|---|---|
| $\phi_{gauss}, \phi^{target} = 1 - 10^{-4}$ | $< 10^{-9}$ | $< 10^{-9}$ |
| $\phi_{gauss}, \phi^{target} = 1 - 10^{-2}$ | $(1.1 \, [0.6 : 1.2]) \times 10^{-3}$ | $(5.3 \, [4.7 : 6.0]) \times 10^{-2}$ |
| $\widetilde{\phi}_{\text{box}}$ | $< 10^{-9}$ | $(2.0 \, [0.7 : 3.0]) \times 10^{-3}$ |

**Table 1.** Results of the transition probabilities for the AMOC model using different score functions

smaller value of $\phi^{target}$ (0.99) or $\widetilde{\phi}_{box}$, we obtain different results: as the shape of the covariance ellipsoid is not spherical
(see Fig. 9), we expect $\phi_{gauss}$ to detect transitions even though the state is actually still far from the destination equilibrium, at
least in certain directions. As a general rule, we expect $\phi_{gauss}$ to give results different from $\phi_{box}$ as long as the ellipsoid of the
system is not spherical, regardless of the choice of the threshold $\phi^{target}$.

## 4  Summary and discussion

We presented and applied several improvements to the TAMS rare-event algorithm, when used to compute transitions in
multistable systems. The first improvement was based on a more rigorous criterion to define noise-induced transitions involving
confidence ellipsoids $\mathcal{E}$ to formalise this criterion. In turn, this led to the rigorous choice of the target set $B = \mathcal{E}_B$ which was
traditionally set by rather arbitrary thresholds. We then showed how to incorporate this definition of $B$ into the score function
$\widetilde{\phi}$. For certain classes of systems, like the ones containing algebraic constraints in addition to differential equations, or when
the noise does not affect one or more directions in the variable space (i.e. the associated covariance matrix is singular), this
method requires some precautions. In particular, for the box model in Castellana et al. (2019), we had to adapt the definition of
the improved score function (19), as well as calculate the pseudo-inverse matrix of the covariance matrix, in order to compute
the ellipsoid.

This method, while being quite general, has several limitations. While the modified score function $\widetilde{\phi}$ is continuous outside
$B$, it is constant in the domain $M = \{\mathbf{X} \in \mathbb{R}^n \mid \phi(\mathbf{X}) > \widetilde{\phi}^{target}\} \setminus \mathcal{E}$ (see Fig. 4). This means that in the TAMS algorithm,
branching will never occur inside $M$, but at the boundary $\partial M$. This can have an influence on the convergence of TAMS if the
level sets of the initial score function $\phi$ have a pathological shape near $\mathbf{X}_B$ and the spatial extension of $D$ is not negligible.
Nevertheless, we expect this to have little impact because this phenomenon is localised near the target state $\mathbf{X}_B$. Therefore,
the trajectories will naturally converge towards $\mathbf{X}_B$ as a result of the dynamics, even without the help of the branching process
of TAMS. However, to ensure that the confidence ellipsoid $\mathcal{E}$ defines a meaningful target set, one needs to be sure that $\mathcal{E}$ is
contained inside the basin of attraction of the target state $\mathbf{X}_B$. While this is the case in the limit of small noise $\sigma \to 0$, it might
not be the case for finite noise. A solution to this issue would be to compute the basin of attraction $\mathcal{V}$ of $\mathbf{X}_B$ and define the
target set $B$ as the intersection $B = \mathcal{E} \cap \mathcal{V}$. However, we expect that in the generic case, this occurs when the noise level $\sigma$ is
high enough so that transitions are less rare and a direct Monte Carlo estimation is sufficient to estimate transition probabilities.





Next, we proposed a systematic method of defining a score function, designed to approximate the static committor, based on empirical transition paths. We proposed an algorithm to estimate the typical transition path under a high noise level, which is then used to define a family of score functions with a single decay parameter $d_0$. We applied our method to a two-dimensional well with a potential barrier. We found that our typical path estimation gave satisfactory results and that the associated score function $\phi_{\mathcal{C}}$, while discontinuous, remained unbiased and relatively insensitive to the change of decay parameter $d_0$. While we did not find significant performance improvements over existing non-trivial score functions, we think that differences will become apparent if applied to higher dimensional systems, where there are more directions to fluctuate in. We did not show any results of the application of this method to the box model of the AMOC (Castellana et al., 2019), because for this rather complicated case, we were not able to efficiently compute typical transition paths from the trajectory histograms of the system.

One key limitation of our approach of constructing the path-based score function $\phi_{\mathcal{C}}$ is the computer memory needed to store the trajectory histogram, which becomes prohibitively huge for high-dimensional systems such as discretised partial differential equation (PDE) systems. As an example, a $50 \times 50$ two-dimensional grid storing 4 variables in each cell (e.g. two velocity components, pressure and a tracer) with 10 bins of resolution in each degree of freedom would require more than 10 Petabytes of memory, which is unfeasonable. However, this limitation can be easily bypassed by defining the objects needed to run the TAMS algorithm, namely the score function $\phi$ and the target set $B$, in a reduced space of much fewer dimensions. For instance, Bouchet et al. (2019) studied the dynamics of the barotropic beta-plane quasi-geostrophic equations describing Jupiter's turbulent atmosphere. While the PDE system was evolved in the full phase space, their rare-event algorithm was run in a reduced 3-dimensional phase space defined by three Fourier coefficients. The target set $B$ and the score function were defined on this reduced space. Moreover, they accumulated transition trajectories in a 3-dimensional histogram and showed their concentration around instantons. By applying the path-finding algorithm, an empirical estimation of the instanton can be made. This offers a viable alternative to solving a minimization problem in the full space to compute the instanton and then project it in the reduced space, which is next to intractable for this system.

Another way to define the reduced space $\mathcal{V}$ in which to run TAMS is to consider the principal components, also called empirical orthogonal functions (EOF), which are the eigenvectors of the covariance matrix. One idea, suggested by Baars (2019), is to retain the EOFs $\{\mathbf{Y}_1^A, \ldots, \mathbf{Y}_k^A, \mathbf{Y}_1^B, \ldots, \mathbf{Y}_{k'}^B\}$ with largest variance (*i.e.* eigenvalue of the covariance matrix) at the initial state $\mathbf{X}_A$ and target state $\mathbf{X}_B$. EOFs represent the directions in which the system fluctuates the most. They are then assumed to be the directions which capture best the noise-driven dynamics. When studying transitions in a PDE model of the Atlantic Meridional Overturning Circulation (den Toom et al., 2011; Baars et al., 2019) projected the dynamics in a reduced space $\mathcal{W} \equiv$ Span$\{\mathbf{X}_A, \mathbf{X}_B, \mathbf{Y}_1^A, \ldots, \mathbf{Y}_k^A, \mathbf{Y}_1^B, \ldots, \mathbf{Y}_{k'}^B\}$ of dimension ($\sim 500$) still too large to apply the histogram method directly. However, one idea is that the TAMS algorithm could be run in an even smaller space $\mathcal{V} \equiv$ Span$\{\mathbf{X}_A, \mathbf{X}_B, \mathbf{Y}_1^A, \ldots, \mathbf{Y}_d^A, \mathbf{Y}_1^B, \ldots, \mathbf{Y}_{d'}^B\}$ (of dimension <10), while still computing the dynamics in the space $\mathcal{W}$. Then, the memory required to store a histogram becomes reasonable and our histogram method can be applied.

Another potential issue of our modified TAMS method is the fact that the score function $\phi_{\mathcal{C}}$ is discontinuous because the trajectory has positive curvature, as shown in Fig. 7(b). The discontinuity is located near the axis $x = 0$. Indeed, when crossing the axis $x = 0$, the closest point on $\mathcal{C}$ changes and $s(\mathbf{X}, \mathcal{C})$ is discontinuous. In fact, in the mathematical proofs about the





statistical and convergence properties of the probability estimator (Cérou et al., 2016), the score functions are assumed to be
continuous. Nevertheless, in our applications, we did not detect any statistically significant bias in the probability estimator
due to the discontinuity. Moreover, some meaning can be attributed to the discontinuity: it is located at the boundary between
the attraction basins of $\mathbf{X}_A$ and $\mathbf{X}_B$ and it thus reflects a qualitative change of behaviour in the system. Crossing this boundary
means that the trajectory converges to $\mathbf{X}_B$ instead of $\mathbf{X}_A$, if $\sigma = 0$. In addition, the remnant of a discontinuity is observed in
the static committor of the similar triple well system used in Rolland and Simonnet (2015). Indeed, their Figure 4c shows the
contour plots of the static committor in the low noise regime. A steep gradient is present at the $x = 0$ boundary, which gives
further evidence that the discontinuity of $\phi_{\mathcal{C}}$ may not be problematic.

Further testing of the ideas presented in this work in high-dimensional systems such as discretised PDEs would give more
insight as to the effectiveness of our approach, compared to the more generic score functions used up to now. Moreover, incor-
porating some form of time-dependence in the score function $\phi$ to specifically optimise TAMS would constitute an interesting
project.

*Code availability.*   The software is available at https://github.com/pascalwangt/PyTAMS and https://github.com/pascalwangt/PyGMAM.





## Appendix A: Trajectory-Adaptive Multilevel sampling (TAMS) algorithm

| | | |
|---|---|---|
| **Input:** | $N$ | number of trajectories in the ensemble |
| | $T$ | total duration of a trajectory |
| | $k_{max}$ | maximum number of iterations of the algorithm |
| | $\phi$ | score function |
| | $\phi_{target}$ | target score, that is the condition that defines the occurrence of a transition |
| **Output:** | $\hat{p}$ | transition probability estimation |

**Initialization:**

1: **for** i=1,…,N **do**

2:      Simulate the trajectory $(\mathbf{X}^{(i)}) = (\mathbf{X}_0^{(i)}, \ldots, \mathbf{X}_T^{(i)})$ of the different ensemble members $i$ (e.g. using the Euler-Maruyama scheme).

3:      Compute the score $\phi_i \equiv \max\limits_{0 \le t \le T} \phi(\mathbf{X}_t^{(i)})$ of trajectory $i$, which is the maximum value of the score function $\phi$ along the trajectory i.

4: **end for**

5: Set the iteration number $k = 1$

**Main loop:**

6: **while** $\min\limits_{1 \le i \le N} \phi_i < \phi_{target}$ and $k < k_{max}$ **do**

7:      Discard the ensemble members for which the trajectories realised the minimum score $i.e$ discard the indices in $S_k \equiv \{j \mid \phi_j = \min\limits_{1 \le i \le N} \phi_i\}$.

8:      Set $l_k \equiv \mathrm{Card}(S_k)$ the number of discarded trajectories ($S_k = \{1\}$ and $l_k = 1$ in Fig. 2. Note that $S_k$ can have multiple elements because of the time discretization).

9:      **for** $j$ in $S_k$ **do**

10:          Choose uniformly at random a trajectory index $r$ in $\{1,\ldots,N\} \setminus S_k$. This is the trajectory from which the new trajectory will originate from ($r = 2$ in Fig. 2).

11:          Copy the trajectory $(\mathbf{X}^{(r)})$ into the new trajectory $(\widetilde{\mathbf{X}}^{(j)})$ up to the first time the value of the score function is greater than $\phi_j$. In other words, set $(\widetilde{\mathbf{X}}_0^{(j)}, \ldots, \widetilde{\mathbf{X}}_{t_{branch}}^{(j)}) \equiv (\mathbf{X}_0^{(r)}, \ldots \mathbf{X}_{t_{branch}}^{(r)})$ where $t_{branch} \equiv \min(\{0 \le t \le T \mid \phi(\mathbf{X}_t^{(r)}) \ge \phi_j\})$.

12:          Generate the rest of the new trajectory up to time $T$ (green branch in Fig. 2) starting from $\widetilde{\mathbf{X}}_{t_{branch}}^{(j)} \equiv \mathbf{X}_{t_{branch}}^{(r)}$ (orange dot in Fig. 2) .

13:          Replace the discarded trajectory $(\mathbf{X}^{(j)}) \leftarrow (\widetilde{\mathbf{X}}_0^{(j)}, \ldots, \widetilde{\mathbf{X}}_{t_{branch}}^{(j)}, \ldots, \widetilde{\mathbf{X}}_T^{(j)})$.

14:          Update the corresponding score $\phi_j \leftarrow \max\limits_{t_{branch} \le t \le T} \phi(\mathbf{X}_t^{(j)})$ which is greater than the previous score by construction of $t_{branch}$.

15:      **end for**

16:      $k \leftarrow k + 1$

17: **end while**

18: Set $N_B$ the number of trajectories having reached the target set $B$, $i.e$ that have a score greater than $\phi_{target}$.

19: **return**

$$\hat{p} = \frac{N_B}{N} \prod_{i=0}^{k} \left(1 - \frac{l_i}{N}\right)$$





## Appendix B: Box model for the Atlantic Meridional Ocean Circulation

The equations determining the evolution of the AMOC in this model are the salinity budgets of the different boxes, together

with the variation of the volume of the pycnocline, and the salt and volume conservation equations (Castellana et al., 2019):

$$
\frac{\mathrm{d}(V_t S_t)}{\mathrm{d}t} = q_S(\theta(q_S)S_{ts} + \theta(-q_S)S_t) + q_U S_d - \theta(q_N)q_N S_t + r_S(S_{ts} - S_t)
$$
$$
+ r_N(S_n - S_t) + 2E_s S_0,
$$
$$
\frac{\mathrm{d}(V_{ts} S_{ts})}{\mathrm{d}t} = q_{Ek}S_s - q_e S_{ts} - q_S(\theta(q_S)S_{ts} + \theta(-q_S)S_t) + r_S(S_t - S_{ts}),
$$
$$
\frac{\mathrm{d}(V_n S_n)}{\mathrm{d}t} = \theta(q_N)q_N(S_t - S_n) + r_N(S_t - S_n) - (E_s + E_a)S_0,
$$
$$
\frac{\mathrm{d}(V_s S_s)}{\mathrm{d}t} = q_S(\theta(q_S)S_d + \theta(-q_S)S_s) + q_e S_{ts} - q_{Ek}S_s - (E_s - E_a)S_0,
$$
$$
\left(A + \frac{L_{xA}L_y}{2}\right)\frac{\mathrm{d}D}{\mathrm{d}t} = q_U + q_{Ek} - q_e - \theta(q_N)q_N,
$$
$$
S_0 V_0 = V_n S_n + V_d S_d + V_t S_t + V_{ts} S_{ts} + V_s S_s, \tag{B1}
$$

where the function $\theta(x)$ is the Heaviside step function. The transports depend on the variables via the following relations:

$$
q_{Ek} = \frac{\tau L_{xS}}{\rho_0 |f_S|},
$$
$$
q_e = A_{GM}\frac{L_{xA}}{L_y}D,
$$
$$
q_U = \frac{\kappa A}{D},
$$
$$
q_N = \eta\frac{\rho_n - \rho_{ts}}{\rho_0}D^2,
$$
$$
q_S = q_{Ek} - q_e, \tag{B2}
$$

where the density of the generic box $i$ is defined as

$$\rho_i = \rho_0\left(1 - \alpha(T_i - T_0) + \beta(S_i - S_0)\right). \tag{B3}$$

The volumes depend, in turn, on the variable $D$:

$$
V_t = AD,
$$
$$
V_{ts} = \frac{L_{xA}L_y}{2}D,
$$
$$
V_d = V_0 - V_n - V_s - V_t - V_{ts}. \tag{B4}
$$

The reference parameter values are shown in table B1.

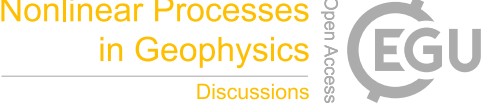

| Parameters used in the model | | |
|---|---|---|
| $V_0$ | $3 \times 10^{17}$ m$^3$ | total volume of the basin |
| $V_n$ | $3 \times 10^{15}$ m$^3$ | volume of the northern box |
| $V_s$ | $9 \times 10^{15}$ m$^3$ | volume of the southern box |
| $A$ | $1 \times 10^{14}$ m$^2$ | horizontal area of the Atlantic pycnocline |
| $L_{xA}$ | $1 \times 10^7$ m | zonal extent of the Atlantic Ocean at its southern end |
| $L_y$ | $1 \times 10^6$ m | meridional extent of the frontal region of the Southern Ocean |
| $L_{xS}$ | $3 \times 10^7$ m | zonal extent of the Southern Ocean |
| $\tau$ | 0.1 N m$^{-2}$ | average zonal wind stress amplitude |
| $A_{GM}$ | 1700 m$^2$s$^{-1}$ | eddy diffusivity |
| $f_S$ | $-10^{-4}$ m$^3$ | Coriolis parameter |
| $\rho_0$ | 1027.5 kg m$^{-3}$ | reference density |
| $\kappa$ | $10^{-5}$ m$^2$s$^{-1}$ | vertical diffusivity |
| $S_0$ | 35 psu | reference salinity |
| $T_0$ | 5 K | reference temperature |
| $T_n$ | 5 K | temperature of the northern box |
| $T_{ts}$ | 10 K | temperature of the box $ts$ |
| $\eta$ | $3 \times 10^4$ m s$^{-1}$ | hydraulic constant |
| $\alpha$ | $2 \times 10^{-4}$ K$^{-1}$ | thermal expansion coefficient |
| $\beta$ | $8 \times 10^{-4}$ psu$^{-1}$ | haline contraction coefficient |
| $r_S$ | $1 \times 10^7$ m$^3$s$^{-1}$ | transport by the southern subtropical gyre |
| $r_N$ | $5 \times 10^6$ m$^3$s$^{-1}$ | transport by the northern subtropical gyre |
| $E_s$ | $0.17 \times 10^6$ m$^3$s$^{-1}$ | symmetric freshwater flux |

**Table B1.** Reference parameters used in equations (B1) - (B4), from (Castellana et al., 2019).

*Author contributions.* PW and DC designed the algorithms, ran the simulations and prepared the figures. All authors discussed the results
and contributed to the writing, of the manuscript.





*Competing interests.* The authors declare that they have no conflict of interest.

*Acknowledgements.* This work was supported by funding from the European Union's Horizon 2020 research and innovation programme for the ITN CRITICS under Grant Agreement Number 643073.

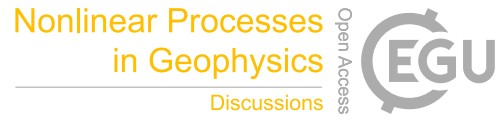

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
