# Peer review of "Improvements to the use of the Trajectory-Adaptive Multilevel Sampling algorithm for the study of rare events"

_Nonlinear Processes in Geophysics, 2020_

## Referee Comment (RC1) · Anonymous Referee #1 · 21 Oct 2020

Review of the manuscript "Improvements to the use of the Trajectory-Adaptive Multilevel Sampling algorithm for the study of rare events".

General comments: The authors conduct their research in a modern and poorly studied field, which is the finite noise induced transition in multi-stable, high-dimensional, non-gradient dynamical systems. The authors are very precise and explicit in defining objectives of the research, in structuring the manuscript and in exposing the obtained results. All sections are clear and well structured. The operation, benefits and limitations of the improved TAMS method has been demonstrated in two typical problems, one of which is two-dimensional double-well system and the other is a box model of

the Atlantic Meridional Ocean Circulation (AMOC).

Specific comments: However, I have a suggestion to make. Due to the complexity of the model in example 2, it was not possible to estimate the typical transition path from the trajectory histograms of the system. Since this technique is one of the main results of the manuscript, it would be beneficial to give another example for multi-stable, high-dimensional, non-gradient dynamical system perturbed by finite noise whose characteristics will allow the application of the method developed.

Please also note the supplement to this comment:
https://npg.copernicus.org/preprints/npg-2020-35/npg-2020-35-RC1-supplement.pdf

———————————————————

---

## Referee Comment (RC2) · Anonymous Referee #2 · 21 Oct 2020

**General comments**

The authors present useful improvements to the Trajectory-Adaptive Multilevel Sampling (TAMS) technique used to study noise induced rare transitions in multi-stable dynamical systems. In particular, the authors focus on developing a mathematical consistent technique for the algorithm to define target sets using confidence ellipsoids around the stable equilibria. Second, the score functions are modified. The benefits of the improved formulation of the algorithm are tested on an exemplary problem and applied to a box model of the Atlantic Meridional Overturning Circulation (AMOC). The ideas are presented clearly, precise and consistent in a well-structured manner. I can

recommend publishing the article after minor revisions.

**Specific comments**

My main concern is that one of the proposed improvements, designing a score function based on a typical transition path which is estimated on transition trajectories on a larger noise level, is not applied to the more complex problem of the AMOC. I understand that the computational memory prohibits a direct estimation. However, the authors present in the discussion that this problem can be handled by reducing the dimensionality of the system (ll. 342-343 & ll. 358-360). So, why not try it with the AMOC box model?

The following specific comments refer to specific parts of the algorithm:

How many starting trajectories are needed for the TAMS to give a robust estimate of transition probabilities?
L.143-144: How can you assure that transitions induced due to large noise follow similar transitions paths as transitions induced by smaller noise levels? How robust is it to use estimates on higher noise levels?
L.195: You state that you estimate the typical transition trajectory on 300 different trajectories. How many trajectories do you need in order to obtain a robust estimate of the typical transition trajectory of a dynamical system?

**Technical corrections**

Figs. 5-8: Panel (b) has only empty parenthesis as label
Figs. 5,7: y-axis has no label
Fig. 8: axes labels are weird in both panels, panel (a) y-tick $10^{-18}$ is not displayed, panel(a) legend box "Monte Carlo" is not displayed.

L. 165-166: "Here we see, which will be more general, that the level sets of the score function do not have the shape of an ellipsoid." - I don't understand the sentence.

[Figure]

L. 241: "Finally, as a main application of one of the techniques shown in this paper..."
- I would suggest to state directly which "one" technique is applied. I know that it becomes clear later in the section, however, I think there is no harm in directly stating what technique is applied.

L. 249: closing parenthesis after $S_s$?

---

## Author Comment (AC1) · 17 Nov 2020

Response to the referees (a formatted .pdf is available in the supplements)

We would like to thank the referees for their time spent to analyze the manuscript, their positive comments, as well as their suggestions to improve the manuscript. Below is our response to the issues raised in the review reports.

Referee #1

General comments

1) The authors conduct their research in a modern and poorly studied field, which is

the finite noise induced transition in multi-stable, high-dimensional, non-gradient dynamical systems. The authors are very precise and explicit in defining objectives of the research, in structuring the manuscript and in exposing the obtained results. All sections are clear and well structured. The operation, benefits and limitations of the improved TAMS method has been demonstrated in two typical problems, one of which is two-dimensional double-well system and the other is a box model of the Atlantic Meridional Ocean Circulation (AMOC).

Response : We would like to thank the reviewer for these positive comments. Changes in text: None

Specific comments

2) However, I have a suggestion to make. Due to the complexity of the model in example 2, it was not possible to estimate the typical transition path from the trajectory histograms of the system. Since this technique is one of the main results of the manuscript, it would be beneficial to give another example for multi-stable, high-dimensional, non-gradient dynamical system perturbed by finite noise whose characteristics will allow the application of the method developed.

Response: We thank the reviewer for the suggestion. The application to a different, and higher dimensional model would be interesting but is outside the scope of this paper as it does not only involve presenting the new model, but also the specific reduction method. Changes in text: None

Referee #2

General comments

1) The authors present useful improvements to the Trajectory-Adaptive Multilevel Sampling (TAMS) technique used to study noise induced rare transitions in multi-stable dynamical systems. In particular, the authors focus on developing a mathematical consistent technique for the algorithm to define target sets using confidence ellipsoids

around the stable equilibria. Second, the score functions are modified. The benefits of the improved formulation of the algorithm are tested on an exemplary problem and applied to a box model of the Atlantic Meridional Overturning Circulation (AMOC). The ideas are presented clearly, precise and consistent in a well-structured manner. I can recommend publishing the article after minor revisions

Response: We would like to thank the reviewer for this positive evaluation. Changes in text: None

Specific comments

2) My main concern is that one of the proposed improvements, designing a score function based on a typical transition path which is estimated on transition trajectories on a larger noise level, is not applied to the more complex problem of the AMOC. I understand that the computational memory prohibits a direct estimation. However, the authors present in the discussion that this problem can be handled by reducing the dimensionality of the system (ll. 342-343 & ll. 358-360). So, why not try it with the AMOC box model?

Response: The application of the histogram method to the AMOC box model is not that interesting as the phase space landscape is relatively simple, and basically comparable to a double-well potential. The application to a different and higher dimensional model would be interesting but is outside the scope of this paper as it does not only involve presenting the new model, but also the specific reduction method. Changes in text: None

3) How many starting trajectories are needed for the TAMS to give a robust estimate of transition probabilities?

Response: The amount of starting trajectories needed to give a robust estimate of transition probabilities depends on the score function. While any score function leads to an unbiased estimation, the variance of the estimator and thus the robustness of the

estimation depend on the score function and the number of starting trajectories. The better the score function performs, the fewer starting trajectories are needed to lower the variance of the estimator. More pragmatically, we ensured that we used enough trajectories, typically one hundred in both models, so that the interquartile range (of the independent realizations) spans less than one order of magnitude around the mean (at maximum 1.5 for the lowest noise point of Fig. 8a). Changes in text: This will be mentioned in the revised section 3.2.

4) L.143-144: How can you assure that transitions induced due to large noise follow similar transitions paths as transitions induced by smaller noise levels? How robust is it to use estimates on higher noise levels?

Response: In a general system, we cannot assure that transitions induced due to large noise follow similar transitions paths as transitions induced by smaller noise levels. However, we can check a posteriori if this is the case by starting at high noise and gradually decrease the noise level. We then apply the empirical estimation of the typical path (or simply compare histograms) each time the noise level is decreased and can detect such change in typical transition paths. High noise estimates are robust as long as there is no drastic change in the behavior of the system at an intermediate noise level, which would signal a physical phase transition in the system. Changes in text: A remark on this will be included in the revised section 2.4 where we also refer to the Rolland and Simmonet (2015) paper.

5) L.195: You state that you estimate the typical transition trajectory on 300 different trajectories. How many trajectories do you need in order to obtain a robust estimate of the typical transition trajectory of a dynamical system?

Response: We need enough trajectories so that the histogram gives an accurate representation of all possible transitions paths so that fluctuations along the typical path can be effectively averaged out by the path estimation algorithm. We expect that the lower the noise level, the fewer trajectories are needed. One way to check if there are enough

trajectories would be to check that the histogram near the equilibria is similar to the Gaussian stationary distribution that arises from locally linearized Ornstein-Uhlenbeck dynamics. For example, starting from an equilibrium point and picking any direction, if for the first few histogram cells the histogram value is not decreasing (as would be the case for a Gaussian distribution), then this would mean that too few trajectories were used.

Changes in text: We will include this in the revised section 3.2.

Technical corrections

Figs. 5-8: Panel (b) has only empty parenthesis as label

Changes in text: Will be corrected.

Figs. 5,7: y-axis has no label

Changes in text: Will be corrected.

Fig. 8: axes labels are weird in both panels, panel (a) y-tick 10-18 is not displayed, panel(a) legend box "Monte Carlo" is not displayed.

Changes in text: Will be corrected.

L. 165-166: "Here we see, which will be more general, that the level sets of the score function do not have the shape of an ellipsoid." - I don't understand the sentence.

Changes in text: Sentence will be rewritten.

L. 241: "Finally, as a main application of one of the techniques shown in this paper. . ." - I would suggest to state directly which "one" technique is applied. I know that it becomes clear later in the section, however, I think there is no harm in directly stating what technique is applied. Changes in text: Agreed; will be mentioned. L. 249: closing parenthesis after Ss? Changes in text: Will be corrected.

Please also note the supplement to this comment:
https://npg.copernicus.org/preprints/npg-2020-35/npg-2020-35-AC1-supplement.pdf